# Breaking the Resistance: Photodynamic Therapy in Cancer Stem Cell-Driven Tumorigenesis

**DOI:** 10.3390/pharmaceutics17050559

**Published:** 2025-04-24

**Authors:** Sheeja S. Rajan, J. P. Jose Merlin, Heidi Abrahamse

**Affiliations:** Laser Research Centre, Faculty of Health Sciences, University of Johannesburg, Doornfontein, P.O. Box 17011, Johannesburg 2028, South Africa; jjayaraj@uj.ac.za (J.P.J.M.); habrahamse@uj.ac.za (H.A.)

**Keywords:** cancer, cancer stem cells, photodynamic therapy, photosensitizers, reactive oxygen species

## Abstract

Cancer stem cells (CSCs) are essential for the growth of malignancies because they encourage resistance to cancer therapy and make metastasis and relapse easier. To effectively tackle the obstacles presented by CSCs, novel therapeutic approaches are required. Photodynamic therapy (PDT) is a promising treatment option for cancer cells, which uses light-sensitive medications that are activated by light wavelengths. This review investigates the use of PDT to overcome malignancies driven by CSCs that have innate resistance mechanisms. PDT works by causing tumor cells to accumulate photosensitizers (PSs) selectively. The reactive oxygen species (ROS), which kill cells, are released by these PSs when they are stimulated by light. According to recent developments in PDT, its efficacy may go beyond traditional tumor cells, providing a viable remedy for the resistance shown by CSCs. Researchers want to improve the targeted elimination and selective targeting of CSCs by combining PDT with new PSs and customized delivery systems. Studies emphasize how PDT affects CSCs as well as bulk tumor cells. According to studies, PDT not only limits CSC growth but also modifies their microenvironment, which lowers the possibility of recovery. Additionally, studies are being conducted on the utilization of PDT and immunotherapeutic techniques to improve treatment efficacy and overcome inherent resistance of CSCs. In conclusion, PDT is a viable strategy for treating carcinogenesis driven by CSCs. By applying the most recent advancements in PDT technologies and recognizing how it interacts with CSCs, this treatment has the potential to surpass traditional resistance mechanisms and improve the future of cancer patients. Clinical and preclinical studies highlight that combining PDT with CSC-targeted approaches has the potential to overcome current therapy limitations. Future efforts should focus on clinical validation, optimizing light delivery and PS use, and developing effective combination strategies to target CSCs.

## 1. Introduction

In the past 20 years, cancer has remained one of the most common causes of mortality across a range of nations, genders, and age groups. It is predicted that 10 million people will die from cancer by the year 2023. The three most lethal cancer categories are lung, breast, and prostate cancers [1,2]. Several variables, including genetic abnormalities, environmental effects, inadequate physical activity, diverse lifestyles, improper diets, tobacco use, and excessive drinking, could be contributing to the increased rate of cancer [3]. Currently offered therapies for treating various malignancies at different stages of the disease include radiation therapy, chemotherapy, surgery, or a combination of these [4]. Unfortunately, the currently available medications have a number of serious adverse effects and dosage-limiting toxicity [5]. It is crucial to concentrate on more recent and effective therapeutic approaches that eliminate cancer cells with little or no adverse effects. To enhance the results of current treatments, studies and clinical trials are being conducted on a few innovative modalities, such as immunotherapy, photodynamic therapy (PDT), virotherapy, and sonodynamic therapy. Though they have been used against cancer cells, these new strategies have not paid much attention to particularly targeting cancer stem cells (CSCs) [6,7].

The primary characteristics of CSCs are their ability for self-renewal and generation of non-stem daughter cells, which comprise most malignancies, and their potential for clonal long-term repopulation [8]. Most remarkably, CSCs can enter an inactive state, which is a reversible cell cycle arrest with low basal metabolism. Recent findings indicate that inactivity is a state that is actively maintained, with signaling pathways playing a role in preserving a balanced state [9]. Inactivity enters and endures metabolic stress while maintaining genomic integrity [10]. According to recent research, CSCs can regenerate themselves, producing several lines of cancer cells that might be helpful in the development of a tumor. CSCs are found in all types of tumors. Furthermore, the tumor’s CSCs may cause aggressiveness, proliferation, and treatment resistance [11]. Thus, a unique strategy for limiting the course of the disease and enhancing therapy is to target CSCs and therapeutic resistance. New drugs that target both cancer and CSCs must be developed because the majority of chemotherapeutics on the market today have several harmful side effects and are only effective against large tumor populations [12]. However, PDT offers considerable promise in eliminating CSCs by causing reactive oxygen species (ROS) bursts through photochemical activity processes, despite treatments for cancer, which generate little endogenous ROS via basic biologic mechanisms [13].

One successful and exciting cancer treatment is PDT. Significant removal of tumors can be accomplished by injecting the patient with a PS that is aimed at the tumor and then directly irradiating the tissue with a laser [14]. While the laser light targets the tumor directly, the PS’s space- and time-selective absorption attributes protect normal cells [15]. PDT has been used extensively throughout the world since it was recognized by the Food and Drug Administration in the United States as an innovative method of therapy to treat cancer patients in clinical settings, either by itself or in conjunction with surgery or chemotherapy [16]. Because of the inadequate PS accumulation in tumors and the feeble resistance to tumor dissemination, PDT’s optimal efficacy in treating malignancies has not been demonstrated [17]. An accurate combination of applied research was employed to solve challenges in the area of biomedical research, leading to the creation of nanotechnology, medical technology, and better healthcare management [18]. In nanomedicine, a single nanoparticle can combine many therapeutic methods to produce a combinatorial or synergistic impact [19]. Numerous flexible nanoformulations, such as liposomes, nanoparticles, nanoemulsions, and micelles, have demonstrated great potential for delivery of novel anticancer drugs, considering expanding interest in nanomedicine [20]. By solving the challenges of limited solubility in water, adverse reactions that frequently occur when delivering drugs, and increasing blood flow for improved tumor development, such nanoparticles may substantially simplify the use of medications for cancer treatment [21]. This review examines current advancements and combinatorial strategies, such as immunotherapy, to improve the effectiveness of cancer therapy. It also examines PDT’s effect on CSCs and their microenvironment, as well as its capacity to overcome CSC resistance in malignancies.

## 2. Essentials of Photodynamic Therapy

Surgery, chemotherapy, radiation therapy, and immunotherapy are the most often used treatment techniques for different forms of cancer. Recently, successful applications of small-molecule-based therapy have also been observed [22]. But when it comes to their capacity to target cancer cells precisely, these conventional treatments have significant drawbacks. Chemotherapy is often linked to systemic side effects and an increased risk of cancer recurrence after surgery, while radiation therapy is limited by the overall dose of radiation [23]. Consequently, research has focused on developing highly targeted and selective, safe, effective, and cost-effective alternative therapy options. Furthermore, it may specifically harm the tumorigenic cells, preserving the normal cells and tissues around them. A new non-invasive treatment option for a number of malignancies is PDT, which can replace conventional tumor-ablative therapies [24].

### 2.1. Mechanism of PDT

Light, PSs, and molecules of oxygen are the three main ingredients of PDT (Figure 1). PDT increases the formation of ROS in two distinct ways. PSs transfer electrons to generate free radicals in the type I photodynamic pathway [25]. PSs can transfer energy to ground-state oxygen (^3^O_2_), resulting in the formation of singlet oxygen (^1^O_2_), through a process known as the type II photodynamic pathway [26]. A ground-state PS accelerates into a high-energy electronic state by light absorption [27]. A consistently excited triplet state (T^1^) can be produced by the intersystem crossing and the PS’s excited singlet state [28]. The T^1^ can release excess energy through various mechanisms, such as heat emission or fluorescence. Alternatively, this energy can be transferred to nearby oxygen molecules or other substrates, leading to the formation of ROS and ^1^O_2_ [29]. During PDT, biological components such as lipids, DNA, and proteins react with ROS, such as superoxide anion radical, hydroxyl radical, and hydrogen peroxide, to kill cancer cells by necrosis or apoptosis [30]. Radiation and chemotherapy, on the other hand, primarily kill tumors by immunosuppressive processes that disrupt tumor vasculatures as an anti-angiogenesis impact and activate the host immune system to find and remove all cancerous cells that may still be present [31]. Furthermore, by properly illuminating the malignancy without damaging the healthy tissues around it, PDT can be specifically applied to a particular tumor location. Consequently, PDT is far more beneficial than chemotherapy and radiation, which have harmful side effects [22]. PDT presents a compelling alternative to traditional treatments, offering benefits such as reduced invasiveness, lower toxicity, and improved patient outcomes. Additionally, PDT allows for repeat treatments, enhances quality of life, and minimizes long-term side effects, making it a promising therapeutic option.

### 2.2. Photosensitizers

The first PS that researchers have looked at is hematoporphyrin derivative, which is used to detect cancer and generate fluorescence [22]. Since 1980, researchers have focused on developing advanced PSs to address the limitations of earlier versions, increasing the use of PDT in cancer therapy. However, only a small number of PSs have been tested in clinical studies [32]. Photofrin, Photosan, Photocan, and other derivatives of porphyrins and hematoporphyrin, each with unique monomeric, dimeric, and oligomeric components, are examples of first-generation PSs [33]. Second-generation PSs have better tissue penetration, higher synthesis of oxygen singlet, and higher chemical purity. Based on their quicker body clearance and higher selectivity for cancerous tissue, PSs also have fewer adverse effects [34]. Nowadays, third-generation PSs are widely used in research to address the shortcomings of their predecessors [35]. Nanostructures, natural PSs, tetrapyrrole structures, and synthetic dyes are the four categories of PSs used in PDT based on their source and composition [36].

### 2.3. Light Source

A range of light sources, such as lasers, traditional lamps, and light-emitting diodes (LEDs), are utilized in PDT to initiate the activation of photosensitizing agents, which then trigger the desired therapeutic effect [37]. An optical system is often needed to broaden a light beam in order to irradiate a larger tissue region, and laser lights are often costly. With non-laser sources, the wavelength of light used for tissue irradiation can be adjusted using optical fibers. Ordinary lighting, on the other hand, can generate heat, which PDT has to avoid. Lastly, PDT’s lighting sources were LEDs. Flexible arrays of LEDs are more broadly accessible, less costly, safer, and thermally non-destructive [38]. Effective PDT relies on precise control over light delivery parameters, including fluency, fluency rate, exposure time, and delivery method, which collectively impact treatment outcomes and therapeutic success [39]. Light intensity refers to the cumulative energy density delivered to a specific area, typically measured in J/cm^2^. In contrast, light intensity rate, expressed in W/cm^2^, represents the rate of energy delivery per unit area per second [40]. Laser-based PDT is commonly employed in both surface and interstitial treatments. Lasers offer distinct advantages, including coherent, monochromatic light emission with tunable wavelengths, allowing for optimal matching with specific photosensitizers and delivering high optical power [41].

### 2.4. Limitations of PDT

Additional work is still needed to develop and improve PS. Several issues need to be addressed before it may be used in the clinic in the future. Several unique PSs and PDT-based creative therapies are currently considerably increasing the safety and effectiveness of PDT in cancer treatment [29]. Each therapeutic strategy has its unique benefits and drawbacks. Optimizing the balance between chemotherapy and PDT in combined treatments is crucial for achieving optimal cancer treatment outcomes. Furthermore, enhancing the targeting efficiency of therapeutic agents is essential to improve the efficacy of combination therapies, particularly in antibody-drug conjugate systems [28]. Despite its potential, the clinical application of PDT is hindered by several limitations. Key challenges include incomplete tumor eradication, restricted light penetration, tumor hypoxia, and suboptimal distribution of photosensitizing agents, which collectively impact treatment efficacy and immune response activation [42].

## 3. Cancer Stem Cells and Tumor Resistance

A scientific dispute on the development of tumors began in 1855 when pathologist Rudolph Virchow’s research showed that malignancies originate from preexisting normal cells [43]. Differentiation refers to the reversible process where specialized cells regress to a less specialized state while remaining within their original cellular lineage, essentially reverting to an earlier stage of differentiation [44]. Differentiation is a fundamental biological process that occurs in various physiological contexts, including tissue repair and regeneration. This process enables cells to regain primitive stem-like traits, facilitating the development of CSCs and contributing to oncogenesis (Figure 2) [45].

Because of its significant impact on tumor resistance and progression, the study of the tumor microenvironment (TME) in solid tumors continues to be a topic of great interest and importance. According to histological analysis, it has been divided into multiple components, and the connections between healthy and cancerous tissue seem to have been broken [46]. A dynamic cancer environment could be produced by establishing niches with various cell types dispersed in different tumor areas. In addition to variations in tumor composition, niches and CSCs can exhibit a substantial phenotypic variety due to overlap in signaling pathways and cell interactions, which gets increasingly complex as the tumor grows. To preserve habitats, CSCs use unique transcriptional and epigenetic markers [47]. CSCs can establish a complex network of interactions between tumor and healthy tissues, creating distinct niches that are interconnected through reciprocal relationships. Hypoxia serves as a key driver of stemness regulators, often leading to the formation of specialized niches that support tumor progression [48]. Due to their rapid metabolism and attraction to resources such as glucose, which encourage tumor migration and dissemination while producing hypoxia and necrosis, CSCs are able to survive [49]. Additionally, CSCs assist the formation of new capillaries and the synthesis of angiogenic factors. They are aided by signals and structures from normal tissue, including extracellular matrix (ECM) and cancer-associated fibroblasts (CAF) [46,50].

In cancer biology, CSCs alternate between active proliferation and a dormant state. During periods of dormancy, CSCs lower their metabolic activity, allowing them to survive for extended periods without dividing. However, in response to external signals, CSCs can reactivate and re-enter the cell cycle, regaining their ability to proliferate [51]. This dual behavior presents major obstacles for chemotherapy, as dormant CSCs are resistant to treatment and can often evolve into more resilient forms [52]. This resistance is primarily due to the nature of traditional chemotherapy, which is designed to target rapidly proliferating cells and functions in a cell cycle-specific manner (Figure 2) [53]. However, due to their slow division and frequent presence in the G1 or S phase, CSCs show resistance to many chemotherapeutic drugs, such as cisplatin, taxol, and doxorubicin [54]. For example, elevated levels of Zinc Finger E-Box-Binding Homeobox 2 (ZEB2) enhance the proportion of colorectal CSCs in the G0/G1 phase, contributing to resistance against platinum-based treatments [55]. Identifying quiescent CSCs versus proliferative CSCs is difficult, as there are no distinct surface markers and the genotypic and phenotypic traits often overlap [56]. CD13 has been suggested as a marker for dormant hepatic CSCs, which are known to counteract chemotherapy-induced ROS and DNA damage [57]. Additionally, epigenetic alterations are essential in controlling the dormant state of CSCs. For instance, SET Domain-Containing Protein 4 (SETD4) supports breast CSC quiescence by trimethylating histone H4 at lysine 20, which promotes the formation of heterochromatin [58]. Increased expression of miR-135a decreases methylation at the CG5 site of the Nanog promoter by directly inhibiting DNA Methyltransferase 1 (DNMT1). This, in turn, allows the interaction between SET and MYND Domain Containing 4 (SMYD4) and the unmethylated Nanog promoter, leading to the activation of Nanog expression in Nanog-negative tumor cells, thereby driving the transition of CSCs [59]. Recent studies have highlighted the role of aldehyde dehydrogenase (ALDH) activity in mediating resistance to therapy in multiple cancers, including breast, pancreatic, lung, Ewing’s sarcoma, stomach, glioblastoma, head and neck, ovarian, and colorectal cancers. This resistance has been observed across a range of chemotherapy agents, such as doxorubicin, paclitaxel, gemcitabine, gefitinib, temozolomide, and platinum-based drugs, positioning ALDH as a critical marker for CSC-related drug resistance [60]. Among the 19 members of the ALDH family, ALDH1 is the most commonly associated with cancer stem cells [61]. Studies indicate mitochondrial alterations in CSCs of chronic myelogenous leukemia (CML) compared to normal stem cells. Resistant CSC subpopulations can be identified by higher mitochondrial mass and increased endopeptidase activity [62]. The chemoresistance mechanisms of CSCs constitute an interactive network (Figure 2). Targeting individual components may not eliminate the resistance posed by CSCs. To devise accurate CSC-targeted treatments that enhance sensitivity, further exploration in the domain of resistance is warranted [43].

## 4. Characteristic of CSC

A landmark discovery in 1997 by Dick and Bonnet significantly advanced the field of stem cell research, as they successfully isolated CSCs with a distinct CD34^+^CD38^−^ phenotype, exhibiting robust proliferative potential in acute myeloid leukemia [63]. This groundbreaking finding confirmed the presence of leukemia stem cells, paving the way for the subsequent introduction of the CSC concept in 2001, which revolutionized our understanding of cancer biology [64]. CSCs represent a small, distinct population of tumor cells that exhibit self-renewal capabilities, contributing significantly to oncogenesis and tumor progression, mirroring the regenerative properties of normal stem cells [65]. The concept of CSCs has undergone significant expansion since its inception, with evidence of their presence in various solid tumors. Notably, Al-Hajj’s 2003 study [66] successfully isolated CD44^+^CD24^−/low^ CSCs from breast cancer, which exhibited robust tumorigenic potential in xenograft models. According to their findings, after 12 weeks, 200 of these cells may cause transplanted tumors in recipient mice. Ten thousand normal breast cancer cells are unable to grow into tumors throughout the same culture period [66]. In the same year, Sheila K. Singh made a significant breakthrough by identifying a distinct population of CD133^+^ CSCs across various types of brain cancers [67]. The discovery of these highly potent CSCs across a broad spectrum of cancers, including prostate, colorectal, pancreatic, and nasopharyngeal malignancies, as well as hematologic cancers, has provided robust validation of the CSC hypothesis [68,69,70].

Research by Ginestier et al. revealed that breast cancer exhibiting high aldehyde dehydrogenase (ALDH) activity possessed a unique combination of properties, including tumor formation, self-renewal capability, and ability to recapitulate diverse characteristics of the original tumor [71]. Interestingly, these ALDH-high cells, constituting a minor fraction of the cancer cell population (less than 1%), shared some characteristics with a previously identified subset of breast CSCs displaying a CD44^+^CD24^−/low^ phenotype [71]. Studies have shown that the biomarker CD133 can be used to identify CSC populations in various types of solid tumors, with particular relevance to certain brain cancer subtypes [72]. However, recent research has raised questions about the reliability of CD133 as a definitive marker for identifying and distinguishing CSCs, sparking ongoing debate about its utility [73]. Recent findings have tempered enthusiasm for CD133 as a CSC marker, highlighting inconsistencies and limitations that have led to uncertainty regarding its effectiveness in identifying and isolating CSCs [74]. Furthermore, research has shown that plasma cells displaying the CD138 phenotype are capable of inducing multiple myeloma in SCID-hu mice but fail to generate similar cancers in NOD/SCID mice, highlighting the complexity of modeling multiple myeloma in different immunocompromised mouse strains [75].

## 5. Molecular Mechanisms of CSCs

The behavior and characteristics of CSCs are influenced by several major signaling pathways, including Wnt, Notch, and Hedgehog. These pathways, which comprise unique combinations of ligands, receptors, and transcription factors, regulate the self-renewal and maintenance of CSCs, ultimately controlling the expression of genes critical for their survival and function (Table 1) [76]. Every CSC route has been linked to the cells’ drug resistance, according to observations [77]. The Wnt signaling pathway is a highly conserved and tightly regulated mechanism that plays a pivotal role in development and tissue homeostasis. Through paracrine signaling, Wnt molecules influence the fate of multipotent stem cells across various species. However, aberrant activation of the Wnt pathway is a common feature in many human cancers, contributing to tumorigenesis and cancer progression [78]. A notable example of aberrant Wnt pathway activation is the inactivating mutation of the APC gene, which significantly increases the risk of colorectal cancer, highlighting the detrimental consequences of uncontrolled Wnt signaling [79]. Mutations in components of the Wnt pathway, such as β-catenin, have been implicated in a wide range of cancers, including skin, liver, prostate, and ovarian malignancies. In CSC, mutations in Wnt pathway components serve a similar function as they do in normal somatic stem cells. Moreover, the Wnt pathway plays a crucial role in maintaining the stem cell niche in tissues such as the colon and lungs, highlighting its importance in both normal and cancer stem cell biology. Consequently, cancer is caused by every mutation that triggers this pathway [80]. Cancer that is resistant to chemotherapy often exhibits reduced expression of Wnt pathway inhibitors, such as WIF1, SFRP, and DKK1. Notably, DKK1 has been shown to sensitize cancer cells to alkylating agents, which induce apoptosis, highlighting the potential therapeutic implications of targeting Wnt pathway regulation in chemotherapy-resistant cancers [39]. Additionally, WIF1 has been found to enhance chemosensitivity of prostate cancer to certain anticancer drugs, including paclitaxel and etoposide, suggesting a potential role for WIF1 in improving treatment outcomes [81].

Notch signaling pathways play a crucial role in controlling cellular behavior, proliferation, and survival across various organisms. This pathway involves the enzymatic activity of γ-secretase, which releases the Notch intracellular domain, allowing it to translocate to the nucleus and modulate gene expression, either by activating or repressing transcription [82]. In humans, improper signaling in the Notch system leads to a variety of issues, such as cancer and evolutionary alterations [83]. In adult stem cells, the Notch network naturally contributes to cell fate and division. Notch genes have been found to have a paradoxical role in cancer, acting as both tumor suppressors and oncogenes depending on cancer types and cellular context. This complexity highlights the challenges of understanding Notch signaling in cancer and the need for further investigation [76]. Activation of the Notch signaling pathway has been observed in CSCs derived from numerous normal tissues, as documented in various research studies [84]. CSCs can provide drug resistance by stimulating the Notch pathway, which inhibits apoptosis. For example, the Notch pathway’s γ-secretase stimulates Akt phosphorylation, allowing cells to avoid apoptosis following chemotherapy. These findings suggest that inhibiting γ-secretase enhances colorectal cancer treatment [85]. Hedgehog signaling pathways play a vital role in embryonic development, and although their activity is generally restricted in adults, they are reactivated in certain contexts, such as tissue repair. However, aberrant activation of the Hedgehog pathway has been implicated in the development and progression of various types of human cancers [86]. Research has established a link between the Hedgehog signaling pathway and maintenance of CSCs. For example, studies have shown that the Hedgehog pathway contributes to self-renewal and preservation of CSC populations, such as the CD44^+^CD24^−/low^ subset in breast cancer, through mechanisms involving regulation of key genes such as BMI-1 [87]. Studies have found that individuals with initial colon cancer have a significantly increased risk, approximately 7–8 times higher CD133^+^ cells than normal cells, and CD133^+^ cells derived from metastatic cells had higher Hedgehog pathway activation than non-metastatic cells [88]. The Hedgehog pathway is also necessary for brain CSC division, survival, regeneration, and carcinogenesis. Eventually, increased Hedgehog pathway activity causes GLI1 and GLI2 to become more activated, which in turn causes cancer (Figure 3) [89].

## 6. Role of CSCs in Tumor Progression and Metastasis

CSCs are distinguished by their ability to self-renew and proliferate, ultimately contributing to the aggressive growth and development of tumors. Scientists think that by elucidating the fundamental process of these cancer cells, we can better understand cellular and molecular biology, which makes the explanation for their growth intriguing [90]. The CSC model posits that a small subset of cells, situated at the apex of the tumor hierarchy, possesses a unique ability to self-renew and differentiate, driving tumor initiation and maintenance [91]. Once activated, CSCs can promote tumor development and metastasis, although they can also be dormant for years [46]. Although the mechanisms behind this feature are still poorly understood, a number of intrinsic systems controlling gene expression seem to play a role in the transition to CSCs that promotes the growth of cancer [92]. Certain cytokines that cause quiescence linked to an ERK^low^/p38^high^ signaling state, as well as signals from the microenvironment such as CXCL12 that activate AKT to promote survival or decreased integrin-mediated mitogenic signaling, are examples of biochemical signaling pathways that help sustain such a dormant state [93]. Through this process, cancer cells can better adapt to their environment, especially in unfavorable circumstances, and CSCs can help the growth of cancer by staying safe and inactive for years and being prepared to reawaken when the situation calls for re-establishing the tumor [94]. Cell plasticity poses a significant challenge in cancer treatment, as it enables tumors to evade therapeutic interventions by either failing to respond or developing resistance to repeated treatments [94]. Cell plasticity enables cells to adapt and change their behaviors in response to environmental cues without requiring genetic mutations, allowing for dynamic cellular reprogramming. This technique encourages the growth of cancer by enabling cancer cells to more effectively adapt to their environment [95]. Because of its significant function in solid tumors, which may lead to tumor resistance and progression, TME research continues to be of great interest and importance [96]. Tumor-associated fibroblasts play a pivotal role in shaping the tumor microenvironment (TME) by releasing growth-promoting factors, such as vascular endothelial growth factor and platelet-derived growth factor, which in turn facilitate accelerated tumor expansion [97]. Additionally, the presence of specific molecular cues and cytokines within the surrounding TME can either promote or suppress tumor progression, influencing the growth dynamics of cancer [98].

These alterations generate microenvironmental signals, including the activation of hypoxia-inducible factors that induce epithelial-to-mesenchymal transition (EMT), driving cells to acquire a stem-like state, thereby enriching the CSC population and potentially promoting metastatic spread [99]. Two transcription factors are present. By blocking E-cadherin, Twist and Snail, two EMT regulators, encourage metastasis. Additionally, the β-catenin signaling pathway is connected to their expression [46]. One hundred years ago, the main cause of metastasis was believed to be the union of malignant cells with macrophages. Subsequent research has yielded compelling evidence that cell fusion can contribute to the development and progression of cancer, shedding light on a potential mechanism underlying tumor growth and dissemination [100]. Heterokaryons and synkaryons are the two forms of hybrids that can arise from the process of cell fusion. The formation of heterokaryons results in the creation of bi- or multi-nucleated hybrid cells, where the genetic material from the parental cells remains segregated within distinct nuclei. This phenomenon was initially observed in vitro using murine Erlich ascites cells infected with Sendai virus, which were fused with human HeLa cells [101]. Notably, Notch1 signaling enables CSCs to migrate and invade, even under normoxic conditions, which typically do not favor metastasis, as observed in ovarian cancer, highlighting the pro-metastatic role of Notch1 in CSCs [102]. Research in breast cancer has revealed that dysregulated Notch signaling promotes the activity of CSCs, driving processes that enhance metastatic potential and self-renewal capacity [103]. A recent cross-sectional study made a significant finding by identifying recurrent genetic patterns in CSCs across both early-stage and locally advanced tumors. Notably, certain genes were found to be consistently expressed in stages I-II, while others emerged in stage IIIA, suggesting their involvement in initial tumor development and subsequent metastasis. Further analysis revealed that expression levels of these genes varied across different stages, implying dynamic changes in gene activity during tumor progression. Additionally, CSCs express many stemness genes more than non-CSCs do, including OCT4, Nanog, and SOX2 [104].

While the behavior of CSCs across different cancer types remains complex and not yet fully understood, there are promising avenues for therapies that specifically target CSCs, EMT, and related signaling pathways. For example, research using a pancreatic cancer model indicates that disrupting stemness can diminish the tumorigenic, metastatic, and recurrent capabilities of CSCs. Targeting key oncogenic pathways such as JAK2/STAT3 and PI3K/mTOR has shown potential as a therapeutic strategy across multiple cancer types [105]. For instance, when paclitaxel was used to treat ovarian cancer cells derived from recurrent patient tumors, it triggered activation of the JAK2/STAT3 signaling pathway and led to elevated expression of CSC-associated genes and proteins in the cells that remained viable [106]. Furthermore, combining paclitaxel with the JAK2-selective inhibitor CYT387 in patient-derived ovarian cancer cells suppressed the paclitaxel-induced activation of the JAK2/STAT3 pathway, as well as the expression of CSC and embryonic stem cell markers in the surviving cells. When these treated cells were introduced into mice, the resulting tumor load was significantly lower compared to that observed in mice implanted with cells exposed to either paclitaxel or CYT387 alone [106]. These preclinical findings have contributed to the advancement of treatments aimed at targeting CSCs. Several signaling pathways associated with CSCs are currently being evaluated in clinical trials across a range of solid tumors and blood cancers, as outlined in Table 2 [107].

## 7. Impact of PDT on CSCs

The generation and elimination of ROS are tightly regulated to maintain a delicate equilibrium, preventing oxidative damage to the organism [108]. Modulating ROS levels presents a promising strategy to disrupt stem cell metabolism, as ROS can significantly influence the cell’s fate, determining whether it proliferates, self-renews, or remains quiescent [109]. PDT triggers a surge in intracellular ROS production, culminating in the excessive oxidation of biomolecules, including proteins, DNA, and lipids. This oxidative stress can directly impact the fate of stem cells. Specifically, ROS-mediated modifications, such as carbonylation, can alter the tertiary structure of proteins, promote protein-DNA and protein-protein cross-linking, and ultimately modulate the activity of key CSC marker proteins, including OCT4 and SOX2 [110]. ROS can directly inflict damage on DNA, leading to point mutations in critical genes, including proto-oncogenes such as Ras and tumor suppressor genes such as p53. Specifically, ROS-mediated oxidation can alter the structure of deoxyguanosine, resulting in mutagenic changes that can occur at a single carbon atom [111,112]. Free radicals can compromise cell membrane integrity by oxidizing lipid peroxides, ultimately triggering ferroptosis, a distinct mode of cell death marked by the accumulation of iron-mediated oxidative damage to lipids [16].

PDT has shown effectiveness in cancer treatment and can contribute to improved patient outcomes. Its benefits are particularly evident when used in combination with other therapies, serving as a primary option for early-stage or precancerous conditions and as a palliative measure in advanced cases. Despite its promise, several limitations continue to hinder the integration of PDT into standard cancer care protocols [113]. A common challenge in PDT is its diminished performance against larger tumor masses, primarily due to the limited ability of the PS or activating light to reach deeper tissue layers effectively. This restriction hampers uniform treatment coverage, reducing the overall therapeutic impact in bulky or deeply situated lesions [114]. In addition to its limited effectiveness in treating larger tumors, PDT is generally unsuitable for addressing metastatic cancer. Metastasis continues to be a major hurdle in oncology, and PDT faces similar limitations. Recurrence is frequently observed in cases where tumor elimination is incomplete, which may result not only from poor tissue penetration but also from the existence of tumor regions that are inherently resistant to PDT [115]. The previously noted limitations of PDT are partly linked to properties of CSCs that enable them to evade this therapy, including their roles in metastasis and their ability to expel drugs via multidrug resistance transporters. Evidence from a study by Morgan et al. (2010) [116] demonstrated that a side population of cells expressing the ATP-dependent transporter ABCG2 contributed to tumor regrowth following PDT. This transporter likely reduces the intracellular concentration of PSs below the lethal threshold, allowing resistant cells to survive and drive tumor repopulation [116].

Similarly to many conventional cancer treatments, PDT also faces resistance from CSCs. Therefore, gaining insight into how CSCs respond to PDT and uncovering the specific mechanisms that allow them to survive and adapt is essential. A deeper understanding of these resistance pathways could pave the way for more effective approaches aimed at fully eliminating CSCs [117]. Table 3 provides a summary of the effects of PDT on CSCs across various tumors, with the detailed findings discussed below. While the impact of PDT on CSCs remains insufficiently explored, several studies have indicated that PDT, on its own, may not be highly effective in targeting CSCs [118]. PDT using polymeric nanoparticles (NPs) co-loaded with TPCS2a and docetaxel in CD44^+^ breast CSCs showed increased PS accumulation and cytotoxicity, especially with prolonged incubation. The treatment reduced cell sphere formation and ALDH1^+^ cell percentage [119]. Breast CSCs were more sensitive to fVII-PDT, showing apoptotic and necrotic cell death. Raschpichler et al. developed an in vitro blood flow model with curcumin-loaded NPs. Laser irradiation reduced cell viability, induced morphological changes, and caused apoptosis and necrosis, suggesting PDT’s potential for metastatic disease treatment [120]. Chizenga et al. (2019) [121] studied aluminum phthalocyanine (AlPcS)-mediated PDT in cervical CSCs, identified by SP and characterized by CD133 and CD49f expression. They found that cellular damage varied with energy density, with lower doses causing blebbing and detachment, while higher doses led to shrunken cells and elevated LDH levels. These results suggest that AlPcS-mediated PDT induces membrane lysis in CSCs [121].

Cogno et al. tested anthraquinone-rich compounds from H. pustulata and T. favicans as PSs in colorectal CSCs. PDT with 5 and 10 J/cm^2^ reduced cell viability by 50% and 75% in monolayer culture. In sphere culture, viability only decreased at higher doses, indicating resistance of colorectal CSCs to PDT [122]. Wei et al. (2014) [123] found that autophagy protects colorectal CSCs from 5-ALA-mediated PDT. PROM1/CD133^+^ CSCs, resistant to PDT, showed increased LC3-II formation and autophagosome activity. Pre-treatment with the autophagy inhibitor chloroquine enhanced PDT cytotoxicity, reduced sphere formation, and decreased CSC tumorigenic potential [123]. A study found that PDT with polyhematoporphyrin (PHP) alone had no effect on nasopharyngeal CSCs. However, combining PDT with 0.5 or 5 µM lovastatin increased PS accumulation and inhibited sphere formation [124]. Oral CSCs with high CD44 expression showed increased miR-145 after 5-ALA-mediated PDT, which enhanced PDT sensitivity, reduced CD44^+^ cells, and decreased sphere formation and invasion [125]. Wang et al. found that glioma CSCs were resistant to 5-ALA-mediated PDT, showing lower PpIX accumulation. This resistance was not due to ABCG2 efflux but was linked to heme oxygenase-1 upregulation, which promotes PpIX degradation. Combining PDT with an iron chelator improved PpIX uptake in vitro [126]. Brain CSCs treated with 5-ALA-mediated PDT showed increased mRNA expression of heme biosynthetic genes [127]. In a study on pancreatic CSCs, 5-ALA-mediated PDT showed that ABCG2 expression inversely correlated with PpIX accumulation. ABCG2 knockdown enhanced PpIX levels and reduced the CSC fraction but did not affect sphere formation [128].

## 8. Selective Targeting of CSCs by PDT

It is noteworthy that during photochemical internalization, PS is modified by biomarkers specific to CSCs, which initially bind to the cell membrane and are then internalized into vesicles. Subsequent PDT releases the internalized molecules, allowing them to exert their effects. While PDT plays a supporting role, photochemical internalization represents a unique strategy for targeting CSCs [129]. Another key enzyme regulated by Wnt signaling is pyruvate dehydrogenase kinase, which inhibits mitochondrial respiration and promotes a shift towards glycolytic metabolism [130]. The Wnt pathway may thus simultaneously regulate CSCs through these two mechanisms. Interestingly, studies have implicated the stem cell marker CD44 in governing glycolytic metabolism, yet the relationship between traditional stemness markers and the metabolic regulation of CSCs remains poorly understood [131]. Researchers discovered that overexpressing RAB5/7 effectively inhibits CSCs. Furthermore, they found that mefloquine hydrochloride modulates RAB5/7 through the endolysosomal pathway, an effect dependent on the PTEN-induced putative kinase 1 and Parkinson disease protein 2 pathway. Although the exact mechanism is unclear, ROS generated by NADPH oxidase may facilitate the interaction between lysosomes and mitochondria [16]. Lysosome-targeting PDT can trigger the release of numerous proteases during photooxidative stress, leading to the activation of endogenous apoptosis-related proteins. Redox-active iron (Fe^2+^) plays a crucial role in facilitating communication between lysosomes and mitochondria. Autophagy involves the sequestration of cytoplasmic material in autophagic vesicles, which are then degraded by lysosomal enzymes, resulting in the accumulation of Fe^2+^ within the lysosomal compartment [132]. PDT also offers distinct advantages through immunogenic activation. Various agents, including mitoxantrone, mitomycin C, and cisplatin, as well as UV light and γ-radiation, can induce endoplasmic reticulum stress (ERS) through ROS-mediated mechanisms. However, this approach can be inconsistent and non-specific. Notably, the pre-apoptotic phase of ERS-mediated immunogenic apoptosis is accompanied by the release of extracellular ATP and calreticulin (ecto-CRT) [133,134]. Research has shown that Photofrin-based PDT primarily induces mitochondrial photooxidative stress, with a minor component of ERS. In contrast, hypericin-mediated PDT (Hyp-PDT) specifically targets the ER, triggering photooxidative stress that can lead to immunogenic apoptosis [135]. Cells targeted by ERS display elevated levels of pre-apoptotic ecto-CRT expression on their surface, surpassing those treated with Hyp-PDT. These findings suggest that PSs targeting the endoplasmic reticulum may be more effective in inducing immunogenic cell death [136].

The WNT/β-catenin, Notch, and Hedgehog pathways are among the key signaling mechanisms that regulate self-renewal in CSCs [137]. Therefore, targeting signaling pathways through various strategies and regulatory levels could enhance the elimination of CSCs. In one study, Karandish et al. used iRGD-functionalized polymersomes to deliver napabucasin, an inhibitor of cancer stemness [138]. The iRGD-targeted polymersomes delivering napabucasin reduced viability in prostate and pancreatic CSCs and downregulated stemness markers such as Notch-1 and Nanog, demonstrating their inhibitory effect. Similarly, Liu et al. developed NPU0126 NPs using PEG-PLA to deliver the MAPK inhibitor U0126 for treating hepatocellular carcinoma. This approach showed strong therapeutic efficacy, reduced toxicity, and significantly impaired sphere formation and CD133^+^ CSC populations, indicating disrupted self-renewal and stemness [139]. To enhance CSC targeting, Miller-Kleinhenz et al. designed ultrasmall magnetic iron oxide NPs (IONPs) functionalized with iWnt and ATF24 peptides targeting uPAR. The iWnt-ATF24-IONP loaded with doxorubicin (DOX) effectively suppressed WNT/β-catenin signaling, reduced uPAR and CSC marker expression, and inhibited invasion and proliferation in CD44^high^/CD24^low^ cancer stem-like cells [140]. Table 4 presents various drug delivery platforms and carriers designed to target CSC-related signaling pathways [138,139,140,141,142,143,144,145,146,147,148,149,150,151].

## 9. Clinical and Preclinical Studies

The therapeutic potential of PDT in cancer treatment was first explored in the late 1970s, with a pioneering study investigating the effect of hematoporphyrin derivative activated by light in five patients with bladder cancer [152]. Since the inaugural study, over 250 clinical trials have been completed, demonstrating the efficacy of PDT. Recent systematic reviews confirm that PDT is a promising therapeutic approach for managing both premalignant and malignant skin cancers [153]. Additionally, PDT can be utilized to irradiate tumor beds and enhance long-term local disease management when combined with surgery. Additionally, PDT has demonstrated outstanding results in numerous clinical trials for a range of upper gastrointestinal malignancies [154]. PDT has been explored as a treatment for malignant brain tumors in various clinical trials, although the majority of these studies were preliminary phase I and II investigations [155]. Assessing the efficacy of PDT is complicated due to the heterogeneity of techniques, adjuvant medications, and tumor subtypes employed across various studies. Moreover, PDT offers potential for tumor irradiation and long-term local disease control, making it a promising therapeutic approach [152].

In a preclinical investigation, mice are inoculated with 350,000 CT-26 cells subcutaneously and subjected to vascular PDT using redaporfin. The treatment regimen consisted of administering 0.75 mg/kg of redaporfin, followed by exposure to 748 nm light at a fluency of 50 J/cm^2^. This experimental setup enabled the assessment of immunological responses elicited by vascular redaporfin-PDT [156]. Foscan^®^ is now being used in a number of preclinical and clinical investigations to treat prostate cancer. In vitro experiments utilizing LNCaP human prostate cancer cells evaluated the photodynamic properties of FosPeg^®^, a liposomal formulation of mTHPC conjugated with polyethylene glycol (PEG) [157]. Although 5-aminolevulinic acid (5-ALA) has been approved for treating basal cell carcinoma (BCC), its use for prostate cancer is still under investigation. Ongoing preclinical and clinical trials are exploring the potential of 5-ALA and its derivatives as therapeutic agents for prostate cancer [158]. Preclinical investigations showed that padoporfin-mediated PDT is a promising, minimally invasive treatment approach for localized, advanced small-cell prostate cancer, offering efficacy and acceptable tolerability [159]. Despite ongoing preclinical and clinical research, no photosensitizers have been approved specifically for breast cancer treatment. Various photosensitizing agents, including “Photofrin^®^, Foscan^®^, Laserphyrin^®^, Purlytin^®^, Verteporfin^®^, and LuTex^®^”, are being investigated in preclinical and clinical trials for their potential in treating breast cancer [160]. The current clinical trials for PDT cancer treatment are displayed in Table 5.

## 10. Cost-Effectiveness and Scalability of PDT Technologies

PDT demonstrated greater health benefits and lower costs (cost/LYS) compared to four cycles of palliative chemotherapy and extensive palliative surgery, making it the more cost-effective option [161]. At a GBP 30,000/LYS threshold, PDT needed to extend life by 61 days over no treatment to be cost-effective. Due to its lower cost, PDT was already cost-effective compared to palliative chemotherapy and surgery. Model data showed PDT adds 129 days over no treatment and 48 days over chemotherapy [162]. Trials selecting patients by tumor size and location for PDT have shown higher survival rates than those reported in this analysis [161]. Four cycles of palliative chemotherapy cost more (GBP 9924) than PDT and no treatment but less than palliative surgery. It also offers a documented survival benefit of 81 days over no treatment [163]. Palliative chemotherapy yields an incremental cost of GBP 41,477 per LYS, GBP 28,764 per overall response, and GBP 153,408 per remission compared to no treatment. At the GBP 30,000 threshold, it is cost-effective only in terms of overall response, not LYS or remission [161]. FullMonteWeb is an open-source, web-based platform for interstitial PDT planning that enables Monte Carlo light simulations and optimization through a user-friendly interface. Hosted on AWS, it offers scalable computing without the need for complex local setup [164].

## 11. Conclusions and Future Perspectives

PDT offers a promising strategy for combating CSCs, which play a major role in tumor formation, metastasis, and therapy resistance. PDT particularly targets CSCs and their microenvironment, inducing direct lethal effects through ROS generation while also having the potential to modify the tumor ecosystem, reducing the likelihood of recurrence. The combination of enhanced PSs and unique delivery systems can improve the selectivity and efficacy of PDT, making it a feasible option for treating malignancies with intrinsic resistance mechanisms. Future research in this sector should focus on refining PDT procedures to maximize therapeutic effects. Research should aim at generating tailored therapy regimens that consider the variety of CSC populations inside cancers. Furthermore, combining PDT with immunotherapeutic methods may increase therapy efficacy by activating immune responses while also targeting CSCs. Understanding the larger ramifications of PDT on cancer biology will require an investigation into the molecular pathways involved. Collaborative studies that investigate the interactions of PDT, CSCs, and the TME may reveal new biomarkers for therapeutic response and resistance. Finally, further innovation in PDT approaches, as well as their incorporation into comprehensive plans for treating cancer, has the capacity to significantly enhance patient outcomes and establish an entirely novel model in cancer therapy.

## Figures and Tables

**Figure 1 pharmaceutics-17-00559-f001:**
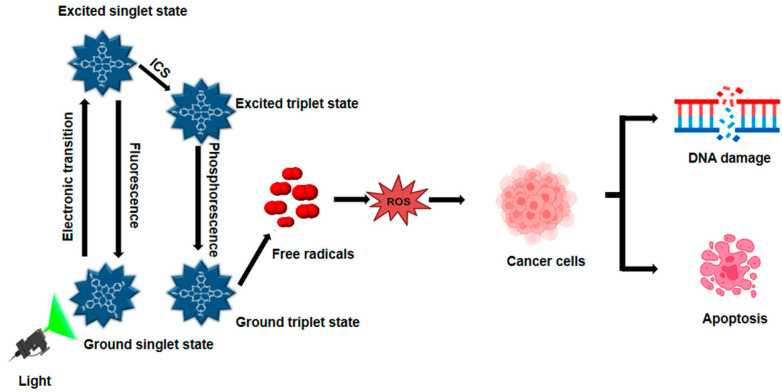
Mechanism of photodynamic therapy. When exposed to light matching its absorption spectrum, a PS transitions from its ground state to an excited state, then to a therapeutic triplet state. This process generates ROS, which induces DNA damage in cancer cells during PDT.

**Figure 2 pharmaceutics-17-00559-f002:**
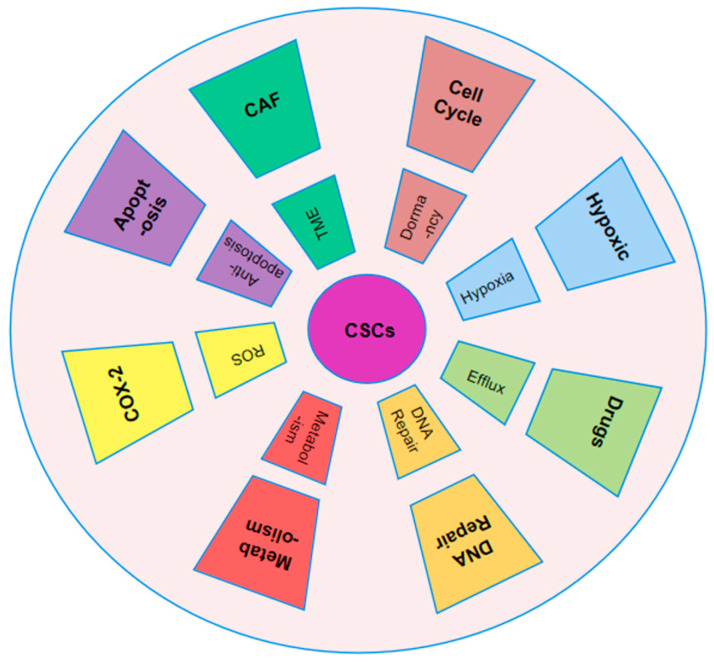
Mechanism of resistance of CSCs to chemotherapy. CSCs resist chemotherapy by remaining quiescent, reprogramming metabolism, using ABC transporters, and activating DNA repair. Their survival is also supported by the tumor microenvironment, including ROS balance, apoptotic signaling, and exosomes from tumor-associated fibroblasts.

**Figure 3 pharmaceutics-17-00559-f003:**
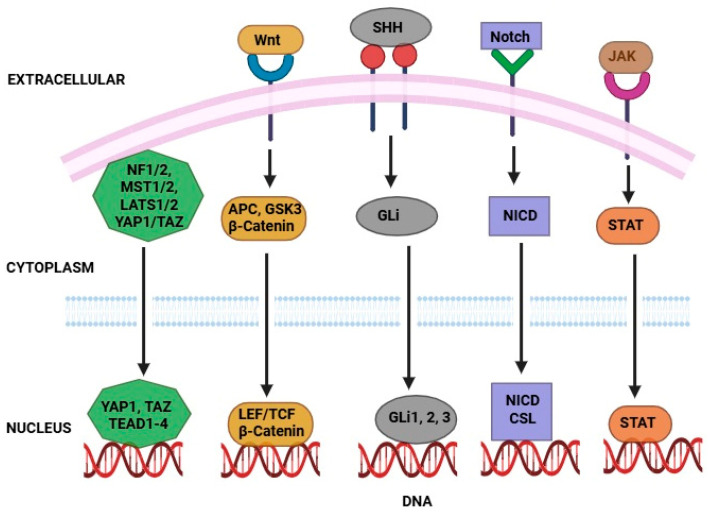
Signaling pathways that regulate CSCs to therapy resistance. Activates genes for CSC maintenance via the β-catenin/TCF complex. NICD enters the nucleus to promote CSC survival and resistance. SHH activates GLI transcription factors, enhancing stemness. YAP/TAZ nuclear translocation drives proliferation and drug resistance when deregulated. STAT dimers induce gene expression supporting CSC traits after receptor activation.

**Table 1 pharmaceutics-17-00559-t001:** Cancer stem cell pathways in cancers.

Cancer Type	Pathway	Function	Therapeutic Agents
Colorectal, breast, skin, central nervous system, prostate, and pancreatic	Notch	Organ development and stem cell differentiation	RO4929097, tarextumab, MK-0752, and demcizumab
Pancreatic ductal adenocarcinoma, glioblastoma, colorectal, medulloblastoma, basal cell carcinoma, and rhabdomyosarcoma	Hedgehog	Repair EMT phenotype, adult stem cell regulation, and maintenance of tissue	Vismodegib, cyclopamine, and sonidegib
Colorectal, gastric, melanoma, skin, pancreatic, breast, and melanoma	Wnt/ β-catenin	EMT phenotype and stem cell self-renewal	LGK974, vantictumab, Ipafricept, PRI-724, OTSA 101, and Foxy-5
Colorectal, breast, melanoma, gastric, head and neck cancer, gastric, hematologic, gynecologic, squamous cell carcinoma, fibrosarcoma, thoracic, and genitourinary	NF-κB	Proliferation, differentiation, and inhibition of embryonic stem cells; regulator activity, immune and inflammatory responses	HGS1029, LCL161, and GDC-0152
Pancreatic ductal adenocarcinoma	JAK/STAT	Self-renewal property in regulation of neurogenesis and hematopoiesis	Napabucasin, pacritinib, and fedratinib
Leukemia, colorectal cancer, glioblastoma, breast cancer, and myeloproliferative disease	PI3K/Akt	Embryonic stem cell regulation, self-renewal, EMT phenotype, intestinal and neuronal stem cells, hematopoietic	MK-2206, idelalisib, temsirolimus, AZD5363, everolimus, dactolisib, and ipatasertib

**Table 2 pharmaceutics-17-00559-t002:** CSC signaling pathways under phase I/II through phase III clinical study.

Cancer Type	Targeted Signal Pathway	Phase
Colorectal	JAK/STAT, Wnt/β-catenin, Nanog	II, III
Gastric/esophageal	JAK/STAT, Wnt/β-catenin, Nanog	III
Gastrointestinal	JAK/STAT, Wnt/β-catenin, Nanog	II
Glioblastoma	JAK/STAT, Wnt/β-catenin, Nanog	Ib/II
Hematological	EphA3, IL-3R	I/II
Hepatocellular carcinoma	JAK/STAT, Wnt/β-catenin, Nanog	Ib/II, II
Mesothelioma	JAK/STAT, Wnt/β-catenin, Nanog	I/II
Mesothelioma	FAK	II
Multiple tumor types	JAK/STAT, Wnt/β-catenin, Nanog	Ib/II
Non-small cell lung	Notch	II
Non-small cell lung	FAK	II
Ovarian	Notch	Ib/II
Ovarian	JAK/STAT, Wnt/β-catenin, Nanog	II
Pancreatic	Notch	II
Small cell lung	Notch	Ib/II, II
Urologic	JAK/STAT, Wnt/β-catenin, Nanog	II

**Table 3 pharmaceutics-17-00559-t003:** Effects of PDT on CSCs in different solid tumors.

Cancer Type	Photosensitizer	Wavelength (nm)	CSCs Isolation Strategies	Observation	Ref.
Breast	TPCS_2a_	488	CD44 expression	Docetaxel (DTX) was ineffective in CSCs, but TPCS_2a_ PDT + DTX was cytotoxic.	[119]
Breast	CUR-NP	447	Metastatic tumor cells	Curcumin-PDT effectively targeted circulating CSCs, inducing apoptosis and early necrosis.	[120]
Cervical	AlPcSmix	673.2	CD133 and CD49f expression	AlPcSmix PDT was dose-dependent, impairing proliferation and inducing necrosis.	[121]
Colorectal	Anthraquinones	420	Sphere formation	Parietin-PDT induced apoptosis, while soranjidol-PDT caused necrosis in spheres.	[122]
Colorectal	Protoporphyrin IX	633	CD133 and PROM1 expression	Autophagy inhibition boosted cytotoxicity and impaired sphere/tumor formation in vitro and in vivo.	[123]
Head and neck	C_34_H_38_N_4_N_a_O_s_	630	CD44 expression	Polyhematoporphyrin-PDT was not cytotoxic to CSC-like cells, but PDT + lovastatin increased cytotoxicity and impaired sphere formation.	[124]
Head and neck	5-ALA	635	Sphere formation CD44 expression	Increased miR-145 expression after 5-ALA-PDT impaired sphere formation and invasion.	[125]
Brain	5-ALA	488	Side population ABCG2 activity	CSCs showed increased HO-1 expression, with PDT resistance not mediated by ABCG2.	[126]
Brain	5-ALA	488	CD133 and Sox2 expression	CSCs showed resistance to ALA-PDT, cisplatin, paclitaxel, irinotecan, and temozolomide.	[127]
Pancreas	5-ALA	488	Side population ABCG2 expression	ABCG2 knockdown improved PpIX accumulation and reduced side population but did not affect sphere formation.	[128]

**Table 4 pharmaceutics-17-00559-t004:** Drug delivery systems and carriers targeting CSC signaling pathways.

Cell Line	Drug	Signaling Pathway	Carrier	Ref.
Human prostate stem cells, human pancreatic cancer stem cells	Napabucasin	Stemness	iRGD peptide-decorated, reduction-sensitive polymersomes	[138]
HepG2, Hep3B, SMMC-7721	U0126	MAPK	PEG-PLA NPs	[129]
MDA-MB-231	DOX, Wnt, and ATF24 peptides	WNT	Dual receptor targeted iWnt-ATF24-IONPActive targeting A15-SLNs	[140]
OSCC cells	Niclosamide	STAT3	Active targeting A15-SLNs	[141]
HEK293, LoVo	CGX1321	WNT	Liposome	[142]
MDA-MB-468, HCC1937	BTZ	Proteasome	PEG-b-PLA NPs	[143]
A549, T98G	PEG-coated GNPs and cold plasma	PI3K/AKT	PEG-coated GNPs	[144]
H-357-PEMT	Quinacrine	P53/P21 dependent manner	QAuNPs	[145]
MDA-MB-231	LY364947 and siPlk1 C	TGF-b	Cationic lipid-assisted polymeric NPs	[146]
MCF-7, MDA-MB-231	siAKT2	PI3K	Triblock structured PM based on the combination of PEI with Pluronic amphiphilic copolymers	[147]
CCL-23, UM-SCC-1	CDF	AKT independent pathway	Liposome	[148]
MDA-MB-231	DAPT	Notch	MSN-PEI-GAorg NPs	[149]
MDA-MB-231	FER siRNA	FER	LMWP	[150]
HCT116	LincRNA-P21	β-Catenin	Ad-lnc-P21-MRE	[151]

**Table 5 pharmaceutics-17-00559-t005:** PDT is being tested in cancer trials.

Cancer Type	Photosensitizer	Phase	Estimated Enrollment	ClinicalTrials.gov Identifier
Lung Cancer	Porfimer sodium	I	12	NCT03678350
	Porfimer sodium	I/II	65	NCT03735095
	Porfimer sodium	I	16	NCT04836429
Head and Neck Cancer	ASP-1929	III	275	NCT03769506
	Porfimer sodium	I/II	82	NCT03727061
	5-aminolevulinic acid hydrochloride	II	26	NCT05101798
Non-Melanoma Skin Cancer	5-aminolevulinic acid hydrochloride	II	28	NCT05020912
	5-aminolevulinic acid hydrochloride	III	186	NCT03573401
Prostate Cancer	Tookad^®^ Soluble	II	50	NCT03315754
	Verteporfin	I/II	66	NCT03067051

## Data Availability

This article does not report any original data, and the research described is based on a review of existing literature.

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
