# Peer review of "Breaking the Resistance: Photodynamic Therapy in Cancer Stem Cell-Driven Tumorigenesis"

_pharmaceutics, 2025, doi:10.3390/pharmaceutics17050559_

Round 1
Reviewer 1 Report
Comments and Suggestions for Authors
Dear Author,
The article Breaking the Resistance: Photodynamic Therapy in Cancer Stem Cell-Driven Tumorigenesis presents an interesting and relevant topic. The author has chosen a significant issue by discussing PDT and immunological PDT. However, the review lacks alignment with the title in some areas, and there are both strengths and weaknesses in its structure.
I have the following suggestions for improvement:
-
The abstract lacks quantitativeness and should include more specific data or findings.
-
Section 2, Essentials of Photodynamic Therapy, is too generalized and may not be necessary in the context of this review.
-
Cancer Stem Cells and Tumor Resistance should be expanded and written in greater detail, supported by figures and tables.
-
Molecular Mechanisms of CSCs should be presented more concisely and supplemented with figures.
-
Role of CSCs in Tumor Progression and Metastasis should be elaborated further, with additional figures and tables to enhance clarity.
-
Impact of PDT on CSCs needs to be expanded, with supporting figures and tables.
-
Selective Targeting of CSCs by PDT should be further detailed, with relevant figures and tables.
-
Consider separating Clinical and Preclinical Studies into two distinct sections and incorporating more case studies to demonstrate the objectives and usefulness of the review effectively.
These improvements would enhance the clarity, depth, and impact of the review.
Best regards,
Author Response
Response to the Reviewer
The article Breaking the Resistance: Photodynamic Therapy in Cancer Stem Cell-Driven Tumorigenesis presents an interesting and relevant topic. The author has chosen a significant issue by discussing PDT and immunological PDT. However, the review lacks alignment with the title in some areas, and there are both strengths and weaknesses in its structure.
We sincerely appreciate your insightful comments and suggestions, which have helped enhance the quality of the manuscript.
I have the following suggestions for improvement:
- The abstract lacks quantitativeness and should include more specific data or findings.
Correction: Thank you for your suggestion. We have revised the abstract to incorporate additional findings.
Clinical and preclinical studies highlight that combining PDT with CSC-targeted approaches has the potential to overcome current therapy limitations. Future efforts should focus on clinical validation, optimizing light delivery and PS use, and developing effective combination strategies to target CSCs.
- Section 2, Essentials of Photodynamic Therapy, is too generalized and may not be necessary in the context of this review.
Correction: We chose not to exclude this section, as it contributes to the overall flow of the review.
- Cancer Stem Cells and Tumor Resistanceshould be expanded and written in greater detail, supported by figures and tables.
Correction: We have expanded the section.
In cancer biology, CSCs alternate between active proliferation and a dormant state. During periods of dormancy, CSCs lower their metabolic activity, allowing them to survive for extended periods without dividing. However, in response to external signals, CSCs can reactivate and re-enter the cell cycle, regaining their ability to proliferate [51]. This dual behavior presents major obstacles for chemotherapy, as dormant CSCs are resistant to treatment and can often evolve into more resilient forms [52]. This resistance is primarily due to the nature of traditional chemotherapy, which is designed to target rapidly proliferating cells and functions in a cell cycle-specific manner (Figure 2) [53]. However, due to their slow division and frequent presence in the G1 or S phase, CSCs show resistance to many chemotherapeutic drugs, such as cisplatin, taxol, and doxorubicin [54]. For example, elevated levels of Zinc Finger E-Box-Binding Homeobox 2 (ZEB2) enhance the proportion of colorectal CSCs in the G0/G1 phase, contributing to resistance against platinum-based treatments [55]. Identifying quiescent CSCs versus proliferative CSCs is difficult, as there are no distinct surface markers and the genotypic and phenotypic traits often overlap [56]. CD13 has been suggested as a marker for dormant hepatic CSCs, which are known to counteract chemotherapy-induced ROS and DNA damage [57]. Additionally, epigenetic alterations are essential in controlling the dormant state of CSCs. For instance, SET Domain-Containing Protein 4 (SETD4) supports breast CSC quiescence by trimethylating histone H4 at lysine 20, which promotes the formation of heterochromatin [58]. Increased expression of miR-135a decreases methylation at the CG5 site of the Nanog promoter by directly inhibiting DNA Methyltransferase 1 (DNMT1). This, in turn, allows the interaction between SET and MYND Domain Containing 4 (SMYD4) and the unmethylated Nanog promoter, leading to the activation of Nanog expression in Nanog-negative tumor cells, thereby driving the transition of CSCs [59]. Recent studies have highlighted the role of aldehyde dehydrogenase (ALDH) activity in mediating resistance to therapy in multiple cancers, including breast, pancreatic, lung, Ewing’s sarcoma, stomach, glioblastoma, head and neck, ovarian, and colorectal cancers. This resistance has been observed across a range of chemotherapy agents, such as doxorubicin, paclitaxel, gemcitabine, gefitinib, temozolomide, and platinum-based drugs, positioning ALDH as a critical marker for CSC-related drug resistance [60]. Among the 19 members of the ALDH family, ALDH1 is the most commonly associated with cancer stem cells [61]. Studies indicate mitochondrial alterations in CSCs of chronic myelogenous leukemia (CML) compared to normal stem cells. Resistant CSC subpopulations can be identified by higher mitochondrial mass and increased endopeptidase activity [62]. The chemoresistance mechanisms of CSCs constitute an interactive network (Figure 2). Targeting individual components may not eliminate the resistance posed by CSCs. To devise accurate CSC-targeted treatments that enhance sensitivity, further exploration in the domain of resistance is warranted [43].
- Molecular Mechanisms of CSCsshould be presented more concisely and supplemented with figures.
Correction: The figure has been updated and incorporated accordingly.
- Role of CSCs in Tumor Progression and Metastasisshould be elaborated further, with additional figures and tables to enhance clarity.
Correction: Thank you for your suggestion. The section is elaborated and added a table.
While the behavior of CSCs across different cancer types remains complex and not yet fully understood, there are promising avenues for therapies that specifically target CSCs, EMT, and related signaling pathways. For example, research using a pancreatic cancer model indicates that disrupting stemness can diminish the tumorigenic, metastatic, and recurrent capabilities of CSCs. Targeting key oncogenic pathways such as JAK2/STAT3 and PI3K/mTOR has shown potential as a therapeutic strategy across multiple cancer types [105]. For instance, when paclitaxel was used to treat ovarian cancer cells derived from recurrent patient tumors, it triggered activation of the JAK2/STAT3 signaling pathway and led to elevated expression of CSC-associated genes and proteins in the cells that remained viable [106]. Furthermore, combining paclitaxel with the JAK2-selective inhibitor CYT387 in patient-derived ovarian cancer cells suppressed the paclitaxel-induced activation of the JAK2/STAT3 pathway, as well as the expression of CSC and embryonic stem cell markers in the surviving cells. When these treated cells were introduced into mice, the resulting tumor load was significantly lower compared to that observed in mice implanted with cells exposed to either paclitaxel or CYT387 alone [106]. These preclinical findings have contributed to the advancement of treatments aimed at targeting CSCs. Several signaling pathways associated with CSCs are currently being evaluated in clinical trials across a range of solid tumors and blood cancers, as outlined in Table 2 [107].
- Impact of PDT on CSCsneeds to be expanded, with supporting figures and tables.
Correction: Thank you for your suggestion. The section has been expanded, and a table has been added.
PDT has shown effectiveness in cancer treatment and can contribute to improved patient outcomes. Its benefits are particularly evident when used in combination with other therapies, serving as a primary option for early-stage or precancerous conditions, and as a palliative measure in advanced cases. Despite its promise, several limitations continue to hinder the integration of PDT into standard cancer care protocols [113]. A common challenge in PDT is its diminished performance against larger tumor masses, primarily due to the limited ability of the PS or activating light to reach deeper tissue layers effectively. This restriction hampers uniform treatment coverage, reducing the overall therapeutic impact in bulky or deeply situated lesions [114]. In addition to its limited effectiveness in treating larger tumors, PDT is generally unsuitable for addressing metastatic cancer. Metastasis continues to be a major hurdle in oncology, and PDT faces similar limitations. Recurrence is frequently observed in cases where tumor elimination is incomplete, which may result not only from poor tissue penetration but also from the existence of tumor regions that are inherently resistant to PDT [115]. The previously noted limitations of PDT are partly linked to properties of CSCs that enable them to evade this therapy, including their roles in metastasis and their ability to expel drugs via multidrug resistance transporters. Evidence from a study by Morgan et al. (2010) demonstrated that a side population of cells expressing the ATP-dependent transporter ABCG2 contributed to tumor regrowth following PDT. This transporter likely reduces the intracellular concentration of PSs below the lethal threshold, allowing resistant cells to survive and drive tumor repopulation [116].
Like many conventional cancer treatments, PDT also faces resistance from CSCs. Therefore, gaining insight into how CSCs respond to PDT and uncovering the specific mechanisms that allow them to survive and adapt is essential. A deeper understanding of these resistance pathways could pave the way for more effective approaches aimed at fully eliminating CSCs [117]. Table 3 provides a summary of the effects of PDT on CSCs across various tumors, with the detailed findings discussed below. While the impact of PDT on CSCs remains insufficiently explored, several studies have indicated that PDT, on its own, may not be highly effective in targeting CSCs [118]. PDT using polymeric nanoparticles (NPs) co-loaded with TPCS2a and docetaxel in CD44+ breast CSCs showed increased PS accumulation and cytotoxicity, especially with prolonged incubation. The treatment reduced cell sphere formation and ALDH1+ cell percentage [119]. Breast CSCs were more sensitive to fVII-PDT, showing apoptotic and necrotic cell death. Raschipichler et al. developed an in vitro blood flow model with curcumin-loaded NPs. Laser irradiation reduced cell viability, induced morphological changes, and caused apoptosis and necrosis, suggesting PDT’s potential for metastatic disease treatment [120]. Chizenga et al. (2019) studied aluminum phthalocyanine (AlPcS)-mediated PDT in cervical CSCs, identified by SP and characterized by CD133 and CD49f expression. They found that cellular damage varied with energy density, with lower doses causing blebbing and detachment, while higher doses led to shrunken cells and elevated LDH levels. These results suggest that AlPcS-mediated PDT induces membrane lysis in CSCs [121].
Cogno et al. tested anthraquinone-rich compounds from H. pustulata and T. favicans as PSs in colorectal CSCs. PDT with 5 and 10 J/cm² reduced cell viability by 50% and 75% in monolayer culture. In sphere culture, viability only decreased at higher doses, indicating resistance of colorectal CSCs to PDT [122]. Wei et al. (2014) found that autophagy protects colorectal CSCs from 5-ALA-mediated PDT. PROM1/CD133+ CSCs, resistant to PDT, showed increased LC3-II formation and autophagosome activity. Pre-treatment with the autophagy inhibitor chloroquine enhanced PDT cytotoxicity, reduced sphere formation, and decreased CSC tumorigenic potential [123]. A study found that PDT with polyhematoporphyrin (PHP) alone had no effect on nasopharyngeal CSCs. However, combining PDT with 0.5 or 5 µM lovastatin increased PS accumulation and inhibited sphere formation [124]. Oral CSCs with high CD44 expression showed increased miR-145 after 5-ALA-mediated PDT, which enhanced PDT sensitivity, reduced CD44+ cells, and decreased sphere formation and invasion [125]. Wang et al. found that glioma CSCs were resistant to 5-ALA-mediated PDT, showing lower PpIX accumulation. This resistance was not due to ABCG2 efflux but was linked to heme oxygenase-1 upregulation, which promotes PpIX degradation. Combining PDT with an iron chelator improved PpIX uptake in vitro [126]. Brain CSCs treated with 5-ALA-mediated PDT showed increased mRNA expression of heme biosynthetic genes [127]. In a study on pancreatic CSCs, 5-ALA-mediated PDT showed that ABCG2 expression inversely correlated with PpIX accumulation. ABCG2 knockdown enhanced PpIX levels and reduced the CSC fraction but did not affect sphere formation [128].
- Selective Targeting of CSCs by PDTshould be further detailed, with relevant figures and tables.
Correction: We appreciate your suggestion; the section has been further elaborated, and a corresponding table has been included to support the content.
The WNT/β-catenin, NOTCH, and Hedgehog pathways are among the key signaling mechanisms that regulate self-renewal in CSCs [137]. Therefore, targeting signaling pathways through various strategies and regulatory levels could enhance the elimination of CSCs. In one study, Karandish et al. used iRGD-functionalized polymersomes to deliver napabucasin, an inhibitor of cancer stemness [138]. The iRGD-targeted polymersomes delivering napabucasin reduced viability in prostate and pancreatic CSCs and downregulated stemness markers like NOTCH-1 and NANOG, demonstrating their inhibitory effect. Similarly, Liu et al. developed NPU0126 NPs using PEG-PLA to deliver the MAPK inhibitor U0126 for treating hepatocellular carcinoma. This approach showed strong therapeutic efficacy, reduced toxicity, and significantly impaired sphere formation and CD133+ CSC populations, indicating disrupted self-renewal and stemness [139]. To enhance CSC targeting, Miller-Kleinhenz et al. designed ultrasmall magnetic iron oxide NPs (IONPs) functionalized with iWnt and ATF24 peptides targeting uPAR. The iWnt-ATF24-IONP loaded with doxorubicin (DOX) effectively suppressed WNT/β-catenin signaling, reduced uPAR and CSC marker expression, and inhibited invasion and proliferation in CD44 high/CD24 low cancer stem-like cells [140]. Table 4 presents various drug delivery platforms and carriers designed to target CSC-related signaling pathways [138-151].
- Consider separating Clinical and Preclinical Studiesinto two distinct sections and incorporating more case studies to demonstrate the objectives and usefulness of the review effectively.
Correction: We have not separated the section, but we have incorporated additional case studies to enhance the quality of the manuscript.

Reviewer 2 Report
Comments and Suggestions for Authors
The manuscript "Breaking the resistance: Photodynamic therapy in cancer stem cell-driven tumorigenesis" by S.S. Rajan, J.P.J. Merlin and H. Abrahamse represents a review on photodynamic therapy (PDT) of cancer, its potential against cancer stem cells (CSC) as well as key features of CSC's metabolism and their role in cancer progression.
The review covers two main subjects, namely a) characteristic and essential properties of cancer stem cells, and b) physicochemical basis and applications of PDT, including ongoing clinical trials. These parts are presented well and the relevant literature is cited properly. However, the authors failed to merge these separate parts properly. The role of PDT in specific eradication of CSCs is hardly described. The difference between CSC and other cancer cells from the standpoint of PDT remains somewhat unclear. If such a difference cannot be pointed out in the context of PDT, the review has to be renamed and reorganized. I strongly suggest to pay more attention to this issue.
The advantages of PDT as the method for cancer treatment is just described in common phrases within the manuscript. This method does demonstrate good results in clinical practice, there are a number of comprehensive reviews on PDT in literature. But, keeping in mind the title and the Abstract of present manuscript, the potential of PDT against CSCs and challenges associated with metabolic peculiarities of CSCs should be emphasized. The discussion of concrete approaches to enhanced photosensitizers' delivery in CSC as well as the comparison of sensitivity of CSC and proliferated cells to PDT would improve the present review.
Minor shortcomes:
- Reference 32 (line 134) probably should be changed.
- Table 1 is hard to read; the columns should be separated from each other; the mistyping "elf-renewal" takes place.
- Table 2 should be corrected (photosensitizer should be specified in rows 1-4, 7-8).
I admit publishing the manuscript by S.S. Rajan and co-authors in Pharmaceutics, but after correction rather than in its present form. Potential and application of photodynamic therapy in relation to cancer stem cells have to be discussed appropriately.
Author Response
Response to the Reviewer
The manuscript "Breaking the resistance: Photodynamic therapy in cancer stem cell-driven tumorigenesis" by S.S. Rajan, J.P.J. Merlin and H. Abrahamse represents a review on photodynamic therapy (PDT) of cancer, its potential against cancer stem cells (CSC) as well as key features of CSC's metabolism and their role in cancer progression.
We are grateful for your constructive feedback, which has significantly contributed to improving the clarity and quality of our manuscript.
The review covers two main subjects, namely a) characteristic and essential properties of cancer stem cells, and b) physicochemical basis and applications of PDT, including ongoing clinical trials. These parts are presented well and the relevant literature is cited properly. However, the authors failed to merge these separate parts properly. The role of PDT in specific eradication of CSCs is hardly described. The difference between CSC and other cancer cells from the standpoint of PDT remains somewhat unclear. If such a difference cannot be pointed out in the context of PDT, the review has to be renamed and reorganized. I strongly suggest to pay more attention to this issue.
Correction: The authors appreciate the valuable comments from the reviewer and thank you for your suggestion. The section has been expanded, and a table has been added.
PDT has shown effectiveness in cancer treatment and can contribute to improved patient outcomes. Its benefits are particularly evident when used in combination with other therapies, serving as a primary option for early-stage or precancerous conditions, and as a palliative measure in advanced cases. Despite its promise, several limitations continue to hinder the integration of PDT into standard cancer care protocols [113]. A common challenge in PDT is its diminished performance against larger tumor masses, primarily due to the limited ability of the PS or activating light to reach deeper tissue layers effectively. This restriction hampers uniform treatment coverage, reducing the overall therapeutic impact in bulky or deeply situated lesions [114]. In addition to its limited effectiveness in treating larger tumors, PDT is generally unsuitable for addressing metastatic cancer. Metastasis continues to be a major hurdle in oncology, and PDT faces similar limitations. Recurrence is frequently observed in cases where tumor elimination is incomplete, which may result not only from poor tissue penetration but also from the existence of tumor regions that are inherently resistant to PDT [115]. The previously noted limitations of PDT are partly linked to properties of CSCs that enable them to evade this therapy, including their roles in metastasis and their ability to expel drugs via multidrug resistance transporters. Evidence from a study by Morgan et al. (2010) demonstrated that a side population of cells expressing the ATP-dependent transporter ABCG2 contributed to tumor regrowth following PDT. This transporter likely reduces the intracellular concentration of PSs below the lethal threshold, allowing resistant cells to survive and drive tumor repopulation [116].
Like many conventional cancer treatments, PDT also faces resistance from CSCs. Therefore, gaining insight into how CSCs respond to PDT and uncovering the specific mechanisms that allow them to survive and adapt is essential. A deeper understanding of these resistance pathways could pave the way for more effective approaches aimed at fully eliminating CSCs [117]. Table 3 provides a summary of the effects of PDT on CSCs across various tumors, with the detailed findings discussed below. While the impact of PDT on CSCs remains insufficiently explored, several studies have indicated that PDT, on its own, may not be highly effective in targeting CSCs [118]. PDT using polymeric nanoparticles (NPs) co-loaded with TPCS2a and docetaxel in CD44+ breast CSCs showed increased PS accumulation and cytotoxicity, especially with prolonged incubation. The treatment reduced cell sphere formation and ALDH1+ cell percentage [119]. Breast CSCs were more sensitive to fVII-PDT, showing apoptotic and necrotic cell death. Raschipichler et al. developed an in vitro blood flow model with curcumin-loaded NPs. Laser irradiation reduced cell viability, induced morphological changes, and caused apoptosis and necrosis, suggesting PDT’s potential for metastatic disease treatment [120]. Chizenga et al. (2019) studied aluminum phthalocyanine (AlPcS)-mediated PDT in cervical CSCs, identified by SP and characterized by CD133 and CD49f expression. They found that cellular damage varied with energy density, with lower doses causing blebbing and detachment, while higher doses led to shrunken cells and elevated LDH levels. These results suggest that AlPcS-mediated PDT induces membrane lysis in CSCs [121].
Cogno et al. tested anthraquinone-rich compounds from H. pustulata and T. favicans as PSs in colorectal CSCs. PDT with 5 and 10 J/cm² reduced cell viability by 50% and 75% in monolayer culture. In sphere culture, viability only decreased at higher doses, indicating resistance of colorectal CSCs to PDT [122]. Wei et al. (2014) found that autophagy protects colorectal CSCs from 5-ALA-mediated PDT. PROM1/CD133+ CSCs, resistant to PDT, showed increased LC3-II formation and autophagosome activity. Pre-treatment with the autophagy inhibitor chloroquine enhanced PDT cytotoxicity, reduced sphere formation, and decreased CSC tumorigenic potential [123]. A study found that PDT with polyhematoporphyrin (PHP) alone had no effect on nasopharyngeal CSCs. However, combining PDT with 0.5 or 5 µM lovastatin increased PS accumulation and inhibited sphere formation [124]. Oral CSCs with high CD44 expression showed increased miR-145 after 5-ALA-mediated PDT, which enhanced PDT sensitivity, reduced CD44+ cells, and decreased sphere formation and invasion [125]. Wang et al. found that glioma CSCs were resistant to 5-ALA-mediated PDT, showing lower PpIX accumulation. This resistance was not due to ABCG2 efflux but was linked to heme oxygenase-1 upregulation, which promotes PpIX degradation. Combining PDT with an iron chelator improved PpIX uptake in vitro [126]. Brain CSCs treated with 5-ALA-mediated PDT showed increased mRNA expression of heme biosynthetic genes [127]. In a study on pancreatic CSCs, 5-ALA-mediated PDT showed that ABCG2 expression inversely correlated with PpIX accumulation. ABCG2 knockdown enhanced PpIX levels and reduced the CSC fraction but did not affect sphere formation [128].
The advantages of PDT as the method for cancer treatment is just described in common phrases within the manuscript. This method does demonstrate good results in clinical practice, there are a number of comprehensive reviews on PDT in literature. But, keeping in mind the title and the Abstract of present manuscript, the potential of PDT against CSCs and challenges associated with metabolic peculiarities of CSCs should be emphasized. The discussion of concrete approaches to enhanced photosensitizers' delivery in CSC as well as the comparison of sensitivity of CSC and proliferated cells to PDT would improve the present review.
Minor shortcomes:
- Reference 32 (line 134) probably should be changed.
Correction: The statement has been revised.
Since 1980, researchers have focused on developing advanced PSs to address the limitations of earlier versions, increasing the use of PDT in cancer therapy. However, only a small number of PSs have been tested in clinical studies.
- Table 1 is hard to read; the columns should be separated from each other; the mistyping "elf-renewal" takes place.
Correction: The necessary revisions have been implemented.
- Table 2 should be corrected (photosensitizer should be specified in rows 1-4, 7-8).
Correction: The required changes have been made.

Reviewer 3 Report
Comments and Suggestions for Authors
The article titled "Breaking the Resistance: Photodynamic Therapy in Cancer Stem Cell-Driven Tumorigenesis" by Rajan et al. provides a comprehensive review of the potential of photodynamic therapy (PDT) in targeting cancer stem cells (CSCs), a subpopulation of tumor cells notorious for driving tumorigenesis, metastasis, and therapy resistance. CSCs exhibit self-renewal, differentiation, and quiescence, enabling them to evade traditional therapies like chemotherapy and radiation. Their ability to reprogram metabolism, activate DNA repair mechanisms, and interact with the tumor microenvironment (TME) contributes to therapeutic resistance and tumor recurrence. The advantages of PDT over conventional therapies include spatial and temporal precision, minimizing damage to healthy tissues. Moreover, PDT can modulate the TME, disrupt CSC niches, and enhance immunogenic cell death, addressing CSC-driven resistance. Preclinical studies demonstrate PDT’s efficacy in CSC-rich models, such as prostate and breast cancer xenografts. Clinical trials highlight PDT’s success in treating skin, gastrointestinal, and brain cancers, though CSC-specific applications remain underexplored. Limitations include light penetration depth, tumor hypoxia, and inconsistent PS distribution. The authors advocate for combinatorial approaches (e.g., PDT with immunotherapy or nanotechnology) to amplify CSC targeting and overcome resistance. The review thoroughly synthesizes molecular mechanisms, technological advancements, and clinical data, offering a holistic perspective on PDT’s potential. It can serve as a valuable resource for researchers and clinicians aiming to advance targeted cancer therapies. However, there are some minor points that should be addressed by the authors before publication:
- While preclinical data are promising, clinical evidence of PDT’s efficacy against CSCs remains sparse. The authors should emphasize and discuss it.
- The review could further discuss cost-effectiveness and scalability of emerging PDT technologies.
- The figures and tables must be fixed. The columns in Table 1 are a little bit merged so the text is hardly readable. The captions of figures 1 and 2 are too detailed and can be shortened.
Author Response
Response to the Reviewer
The article titled "Breaking the Resistance: Photodynamic Therapy in Cancer Stem Cell-Driven Tumorigenesis" by Rajan et al. provides a comprehensive review of the potential of photodynamic therapy (PDT) in targeting cancer stem cells (CSCs), a subpopulation of tumor cells notorious for driving tumorigenesis, metastasis, and therapy resistance. CSCs exhibit self-renewal, differentiation, and quiescence, enabling them to evade traditional therapies like chemotherapy and radiation. Their ability to reprogram metabolism, activate DNA repair mechanisms, and interact with the tumor microenvironment (TME) contributes to therapeutic resistance and tumor recurrence. The advantages of PDT over conventional therapies include spatial and temporal precision, minimizing damage to healthy tissues. Moreover, PDT can modulate the TME, disrupt CSC niches, and enhance immunogenic cell death, addressing CSC-driven resistance. Preclinical studies demonstrate PDT’s efficacy in CSC-rich models, such as prostate and breast cancer xenografts. Clinical trials highlight PDT’s success in treating skin, gastrointestinal, and brain cancers, though CSC-specific applications remain underexplored. Limitations include light penetration depth, tumor hypoxia, and inconsistent PS distribution. The authors advocate for combinatorial approaches (e.g., PDT with immunotherapy or nanotechnology) to amplify CSC targeting and overcome resistance. The review thoroughly synthesizes molecular mechanisms, technological advancements, and clinical data, offering a holistic perspective on PDT’s potential. It can serve as a valuable resource for researchers and clinicians aiming to advance targeted cancer therapies. However, there are some minor points that should be addressed by the authors before publication:
We deeply appreciate your valuable feedback, which has played a crucial role in refining and enhancing the overall quality of our manuscript. We have made revisions to all sections.
- While preclinical data are promising, clinical evidence of PDT’s efficacy against CSCs remains sparse. The authors should emphasize and discuss it.
Correction: Thank you for the comment. We have discussed the clinical evidence in Sections 6 and 7.
- The review could further discuss cost-effectiveness and scalability of emerging PDT technologies.
Correction: Thank you for the comment. We have addressed and included a discussion on the cost-effectiveness and scalability of emerging PDT technologies.
- Cost-effectiveness and Scalability of PDT Technologies
PDT demonstrated greater health benefits and lower costs (cost/LYS) compared to four cycles of palliative chemotherapy and extensive palliative surgery, making it the more cost-effective option [161]. At a £30,000/LYS threshold, PDT needed to extend life by 61 days over no treatment to be cost-effective. Due to its lower cost, PDT was already cost-effective compared to palliative chemotherapy and surgery. Model data showed PDT adds 129 days over no treatment and 48 days over chemotherapy. [162]. Trials selecting patients by tumor size and location for PDT have shown higher survival rates than those reported in this analysis [161]. Four cycles of palliative chemotherapy cost more (£9,924) than PDT and no treatment but less than palliative surgery. It also offers a documented survival benefit of 81 days over no treatment [163]. Palliative chemotherapy yields an incremental cost of £41,477 per LYS, £28,764 per overall response, and £153,408 per remission compared to no treatment. At the £30,000 threshold, it is cost-effective only in terms of overall response, not LYS or remission [161]. FullMonteWeb is an open-source, web-based platform for interstitial PDT planning that enables Monte Carlo light simulations and optimization through a user-friendly interface. Hosted on AWS, it offers scalable computing without the need for complex local setup [164].
- The figures and tables must be fixed. The columns in Table 1 are a little bit merged so the text is hardly readable. The captions of figures 1 and 2 are too detailed and can be shortened.
Correction: All requested revisions have been addressed.
Figure 1. Mechanism of photodynamic therapy. When exposed to light matching its absorption spectrum, a PS transitions from its ground state to an excited state, then to a therapeutic triplet state. This process generates ROS, which induce DNA damage in cancer cells during PDT.
Figure 2. Mechanism of resistance of CSCs to chemotherapy. CSCs resist chemotherapy by remaining quiescent, reprogramming metabolism, using ABC transporters, and activating DNA repair. Their survival is also supported by the tumor microenvironment, including ROS balance, apoptotic signaling, and exosomes from tumor-associated fibroblasts.

Round 2
Reviewer 1 Report
Comments and Suggestions for Authors
Dear Author,
I appreciated the comments addressed.
Reviewer 2 Report
Comments and Suggestions for Authors
Authors made necessary additions to the manuscript. I believe that it deserves publishing in Pharmaceutics in its present form.